# Student-Led Motivational Interviewing for Physical Activity Promotion among Rural Adults: A Feasibility and Acceptability Trial

**DOI:** 10.3390/ijerph18031308

**Published:** 2021-02-01

**Authors:** Jenelle Dziano, Emma Milanese, Svetlana Bogomolova, James Dollman

**Affiliations:** 1Allied Health and Human Performance, University of South Australia, Adelaide 5001, Australia; rayjk001@mymail.unisa.edu.au (J.D.); emma.milanese@unisa.edu.au (E.M.); 2Ehrenberg-Bass Institute for Marketing Science, University of South Australia, Adelaide 5001, Australia

**Keywords:** rural, physical activity, health promotion, motivational interviewing

## Abstract

In many countries, rural residents have lower life expectancies and poorer health outcomes than urban residents. Adults living in rural Australia have lower physical activity levels than major city counterparts, contributing to this observed health disparity. As physical activity interventions in rural populations have shown minimal success, there is an urgent need for innovative and affordable interventions that facilitate active lifestyles in this vulnerable population. This study assessed the feasibility of physical activity-focused motivational interviewing, delivered by university health sciences undergraduates in a rural Australian region. “Health age” was assessed at baseline (*n* = 62) from physiological and behavioral measures, immediately followed by the motivational interview, with health age again assessed at 8 weeks follow-up. Mixed methods using a questionnaire (*n* = 41 at both time points) and one-on-one interview (*n* = 8) identified aspects of intervention acceptability and feasibility. A large majority rated the motivational interview as meaningful (98%), empathetic (96%), autonomy-focused (88%), and likely to lead to sustained behavior change (98%). Interviews highlighted several potential attitudinal and structural factors that might influence long-term behavior change. Further development of this strategy in rural regions will depend on a deeper understanding of individuals’ and communities’ awareness, attitudes, and beliefs in relation to active lifestyles.

## 1. Introduction

Physical inactivity is a major public health concern. Insufficient physical activity is regarded by the World Health Organization (WHO) as one of the leading risk factors for morbidity and premature mortality worldwide [1]. Lack of physical activity is a major contributor to obesity, with obesity in turn a major risk factor for type 2 diabetes mellitus, cancer, and cardiovascular disease. Independent of the association with obesity, physical activity can also influence glucose tolerance, hypertension, and hyperlipidemia, all of which are risk factors for chronic diseases [2].

In many countries including Australia, rural residents have lower life expectancies and poorer health outcomes than urban residents [3,4,5], and this disparity is only partly attributable to socioeconomic differences [6]. Notably, the majority of adult Australians fail to achieve levels of physical activity that are associated with health benefits [7], and adults living in rural and remote Australia have even lower physical activity levels than their counterparts in major cities [8]. While there is a common perception that rural residents typically engage in more physically demanding occupations such as farming, only 30% of rural males and 21% of rural females are sufficiently active [8].

Research suggests that the availability of evidence of health benefits is in itself insufficient for the adoption of regular physical activity, and that comprehensive translation mechanisms are needed [9]. The individual, social, and environmental factors that influence physical activity differ between urban and rural adults [10], and therefore, contextually tailored behavioral interventions are needed to promote healthy lifestyles. However, it is currently unclear how best to intervene to promote physical activity and discourage sedentary behaviors among adults living in rural settings. To date, physical activity interventions in rural populations have shown minimal success [11], underscoring the urgent need for innovative, feasible, and contextually appropriate interventions that motivate and facilitate active lifestyles in this vulnerable population [2].

The feasibility and acceptability of an interactive software program, HealthScreenPro [12], that estimates “health age” compared with actual age, as a motivator for adoption of regular physical activity has been previously trialed in rural South Australia [13]. The current study explored participants’ perceptions of the addition of a motivational interview (MI) immediately following health age assessment, to capitalize on elevated motivation to increase physical activity, a so-called “teachable moment”. There is growing interest in MI, based on self-determination theory (SDT), as a prompt to develop active lifestyles [14]. The feasibility of competently delivered MI in rural regions is challenged by relatively low access to suitably qualified health professionals [8]. Accordingly, this single-arm study tested feasibility and acceptability of a physical activity-focused MI and two tailored follow-up text or telephone contacts that reinforced MI content, delivered by clinical placement university health sciences students at low cost, in a rural South Australian region. The significance of the study lies in its potential to lead to achievable, culturally acceptable, sustainable, and efficacious health promotion in typically “hard to reach” rural populations who are currently at particularly elevated risk of several lifestyle-related chronic diseases.

## 2. Materials and Methods

### 2.1. The Intervention

In 2018, a convenience sample of residents in South Australia’s Barossa region was recruited through strategically placed flyers (posted in prominent locations such as health services waiting rooms, shop windows, and community meeting places) and local media, including newspapers and radio. Interested people were asked to contact project staff by telephone, whereby details of the project were provided. Participants initially completed a questionnaire on current physical activity levels [15] and stage of behavior change according to the Transtheoretical model [16]: precontemplation, contemplation, preparation, action, or maintenance. Health age was then estimated from health and behavioral measures, details of which have been reported [13]. In brief, current medications, health status, height, weight, blood pressure, lung function, total cholesterol, blood glucose, postural balance, and self-reported daily physical activity were entered into HealthScreenPro^®^ which generated an overall summary of the participant’s health status as a health age [12].

Informed by questionnaire responses and health measures, a one-on-one MI framed in SDT was conducted by final-year health sciences students, who had received two days of intensive training in the principles and delivery of MI. The participant-driven conversation aimed to promote planning self-efficacy, autonomy, relatedness, and competency to facilitate physical activity behavior change. At four and six weeks follow-up, participants were contacted by phone or text message to reinforce key messages from the MI, tailored to each participant’s stage of behavior change. After the eight-week intervention, participants were again administered questionnaires on physical activity levels [15] and stage of behavior change [16], and underwent a follow-up health age assessment.

### 2.2. Evaluation of the Intervention

A mixed-method evaluation was undertaken to allow a broad range of questions to be addressed and to generate more complete knowledge to inform practice [17]. Specifically, a questionnaire and an in-depth semi-structured interview using a storyboard were used to investigate feasibility and acceptability of the intervention. The questionnaire specifically assessed participants’ experience of the MI and their impressions of delivery fidelity. Through a flexible and dynamic process, the interviews enabled a richer insight into participants’ experience of the intervention as a whole, focusing on factors at each stage that facilitated or hindered engagement and compliance.

#### 2.2.1. Study Participants

All intervention participants (*n* = 62) were eligible for inclusion in the evaluation and were issued an invitation to participate following the baseline health age assessment. Of these, 52 provided formal consent. Interview participants were selected by purposive sampling to ensure a rich description of participant experience. Specifically, change in questionnaire-reported physical activity [15] was trichotomized as “high responders”, “low responders”, and “non-responders”, and three participants from each of these categories were selected and invited, while maximizing representation across age, gender, and health status. Semi-structured in-depth interviews were conducted with eight of the nine invited participants.

#### 2.2.2. Data Collection

##### Questionnaires

The questionnaire, adapted from Sekhon, Cartwright, and Francis [18], asked participants to assess, on a Likert Scale, 10 domains of feasibility and acceptability. Participants completed the questionnaire after the MI and at the follow-up health age assessment.

##### Interviews

In-depth semi-structured interviews were conducted 6–8 weeks post-intervention at a community facility familiar to the participants. Interview duration ranged from 26 to 68 min. The interviews focused on barriers to and enablers of engagement with, and persistence with, the intervention. The interview guide was developed on the basis of previous studies of barriers and enablers of engagement in active lifestyles among rural South Australians [10,19].

##### Data Analysis

Changes in questionnaire responses between baseline and post-intervention were tested using Chi square, with inference of significant change set at *p* ≤ 0.05. Interviews were audio-recorded, transcribed verbatim, and manually coded to analyze content. Interviews were analyzed independently by two researchers (first and second authors), with transcripts read and re-read allowing for synthesis of divergent views. Investigator triangulation was used to overcome the strength and weaknesses and associated biases of a particular approach [19].

Ethics approval was granted by the University of South Australia Human Research Ethics Committee (ID200412).

## 3. Results

A total of 58 participants (39 females and 19 males; average age = 64.0 years) had their health age assessed at baseline (time point 1) with 45 participants returning at time point 2 for health age assessment. Table 1 summarizes health age and individual health variables at both time points.

### 3.1. Questionnaires

Forty-one participants (80% retention; 64.1 years; 67% females) provided useable questionnaire data at both time points. Baseline analysis using t tests revealed that those who completed questionnaires only at baseline did not differ from those who completed questionnaires at both time points, on any of the demographic, behavioral, and health measures. At baseline, the large majority identified the strategy as low burden (85%) and consistent with daily responsibilities and personal values (92%). Most participants viewed the baseline MI as: meaningful (98%); clear in its goals (96%); empathetic (96%); beneficial in relation to the effort invested (94%); autonomy-focused (88%); enhancing of planning self-efficacy (84%); and likely to lead to sustained behavior change (98%). Between time points, there were no differences in response frequencies for any questionnaire item (all *p* > 0.05).

### 3.2. Interviews

A thematic analysis identified the following key themes: accessibility; credibility and clarity of intervention; community social and environmental influences; attitudes and beliefs; and knowledge of the benefits of physical activity. Findings are reported according to their analytical typologies and verbatim quotes are labeled according to participant gender, age, and response category: high responder (HR), low responder (LR), and non-responder (NR).

### 3.3. Accessibility

The intervention was conducted in a large meeting room in a local shopping center overlooking a supermarket. The building provided lift and stair access to the assessment area. Participants generally reported the location as convenient and easy to access. When questioned on proximity of location, ease of parking, and navigation to the area, participants reported no concerns.


*“I thought that the choice of venue was probably very good but anywhere local I think anybody would find it …they wouldn’t blink over anywhere in this region, so travel is not really an issue.”*
(Female, 82, LR)


*“It’s only 20 km yeah its good.”*
(Female, 72, LR)

While no access concerns were noted, one participant did propose a potential access issue for her peers.


*“A lot of women my age or a bit younger don’t drive so they have got to rely on their husbands.”*
(Female, 72, LR)

### 3.4. Clarity and Credibility of Intervention

To inform the MI, prior to the first health age assessment, participants were asked to complete a questionnaire on demographic information and psychological readiness. Participants reported being willing to complete the questionnaires; however, some concerns were raised around the clarity and ambiguity of some questions.


*“I was happy to fill them out, but I struggled with that sort of thing in you know, could be this, could be something else.”*
(Female, 69, NR)


*“…afterwards I thought oh maybe they meant that.”*
(Female, 73, NR)

Some participants questioned the validity of their responses upon later reflection and indicated that they would prefer to receive the questionnaires in advance.


*“I was probably a little bit out because there were a few things I thought about later on because I do more driving than what I thought I did because I sort of added it all up over a month and averaged it and so for daily activity you know I probably sit a bit more than what I thought I did.”*
(Male, 72, HR)


*“It’s easier to think when you are in your own environment.”*
(Female, 66, NR)

One participant questioned the accuracy of the health assessments.


*“I wasn’t sure some of your things were very accurate.”*
(Female, 73, LR)

Despite her concerns, she did note that the results she received were similar to recent tests directed by her GP.

### 3.5. Social Influences

The MIs were conducted one-on-one; however, there were 4–5 concurrent interviews in an open space. Generally, participants were comfortable with this arrangement although one participant expressed concern about confidentiality in this setting.


*“I didn’t feel comfortable was when I was doing the paperwork and two other people were with me and I thought we were going to do it all together in that group.”*
(Female, 39, LR)

When questioned on the level of engagement in the MI, one participant reported being concerned with holding other people up.


*“I was conscious of time and I often feel like whatever I do I’m always the last one to leave. I remember chatting here, I don’t know, with whoever and I thought, ‘oh my god, they’re probably sick of me, I wish this woman would get out of here’.”*
(Female, 73, LR)

One of the team members assisting with health age assessments was from the local community, which appeared to have a positive impact on the participants’ experience.


*“She’s a local girl… I guess that was the first bit that made me feel comfortable.”*
(Female, 69, NR)

### 3.6. Prior Knowledge and Exposure to the Concept of Health Age

Many participants reported being familiar with the health age concept prior to hearing about the intervention. Previous exposure to the concept appeared to be a key driver for participation for many, with one participant proposing community education sessions as a means of increasing interest in this population.


*“Well I have to say that what motivated me to come was that, and I probably mentioned that a program on the ABC and it was an English program, about health, people’s age, their real—what do you call it, their age.”*
(Female, 69, NR)


*“And we had seen it online on Michael Mosley—on a couple of shows on TV so when I saw it down here I thought yeah I wouldn’t mind seeing.”*
(Female, 69, HR)


*“…running something like that (education session) before you did your survey, you might get people on the day enrolling so you might pick up many more participants.”*
(Female, 82, LR)

### 3.7. Attitudes and Beliefs

The intervention aimed to increase physical activity, either by increasing duration, frequency, or intensity of engagement, or a combination of these according to participant preferences. Participants were encouraged to choose activities and intensities that were enjoyable; however, despite this message, participants exhibited a range of persistent beliefs regarding intensity, duration, and mode of physical activity to yield benefit. In some instances, participants’ outdated beliefs about required intensity were reinforced by their health providers.


*“With the Michael Mosley things I’m starting to do like shorter exercise sessions but more intense, like walking but fast walking those kinds of things to increase my heart rate.”*
(Female, 69, HR)

One participant had previously told a diabetic nurse that she parked her car and walked around town and reported that the nurse responded with:


*“Oh but how fast did you walk, you have to walk fast.”*
(Female, 73, LR)

### 3.8. Motivation

Despite rating feasibility and acceptability of the MI high through the questionnaire, some participants identified (lack of) motivation as a continuing barrier.


*“It’s achievable if I could be bothered…It should be important to me but it’s not. I don’t really care.”*
(Female, 72, LR)


*“I really know what to do, I just haven’t got the willpower to do it…I think what was really getting me down was my weight and my lack of energy to get up and do anything. You just felt like, ugh, after I’d done a bit of gardening and things like that.”*
(Female, 69, NR)

### 3.9. Community and Social Influences

The environment and access to fitness facilities were reported barriers to physical activity.


*“I don’t like this idea for the gym you’ve got to go to the doctor, you’ve got to get all these forms signed and stuff, I’m really past all that. I can’t be bothered with that.”*
(Female, 73, LR)


*“It (the footpath) is just bitumen but its full of holes and it’s dangerous.”*
(Female, 39, LR)

Participants identified positive and negative social influences in relation to enacting lifestyle changes.


*“One of the reasons it (Weight Watchers) works for me anyway is going to a meeting and just seeing other people and knowing that people have their ups and downs.”*
(Female, 69, NR)


*“When you go for a care plan if you put on weight, they have a go at you.”*
(Female, 72, LR)


*“I don’t want everyone knowing my weight…I’m not proud of it at the moment.”*
(Female 39, NR)


*“I think they would have been embarrassed to front up or face the reality of a health age compared with your chronological age.”*
(Female, 69, HR)

### 3.10. Knowledge of the Benefits of Physical Activity

Participants exhibited different levels of understanding regarding the physical and psychological benefits of physical activity.


*“In the morning, I get up and I feel ugh ugh, I feel like a ninety-year-old until I get moving and then I’m not too bad.”*
(Female, 69, NR)


*“Having made an effort, just going for walks, a little bit, even just 20 min around the block or whatever. I was doing that and that always starts you off on a good foot.”*
(Female, 69, NR)


*“I didn’t realize exercise is good for diabetes, I haven’t got diabetes but exercise is good for everything. I didn’t realize. I didn’t even think.”*
(Female, 72, LR)

A high-responding participant reported being surprised at the impact on his health age of a small amount of exercise.


*“I really hadn’t changed between the two (assessments) although I did start doing a bit more exercise like weight stuff and things like that, but not a great deal.”*
(Male, 72, HR)

Another high responder acknowledged that the MI had resulted in a positive attitude to regular physical activity.


*“My knowledge now is very good but I didn’t have that knowledge when I was raising children…That’s another learning that I’ve developed from myself as well is that it’s not about beating yourself up over the gym and in actual fact gyms aren’t always the answer for everybody.”*
(Female, 69, HR)

## 4. Discussion

This study examined the experiences of rural participants in a novel MI-based physical activity intervention. By exploring aspects of feasibility and acceptability, themes have been identified that can provide insight into barriers and enablers of engagement with physical activity promotion in a population at relatively high risk of hypokinetic chronic diseases. Questionnaires revealed that participants strongly endorsed feasibility of the MI component of the intervention and its capacity to satisfy needs according to SDT, with these positive perceptions persisting across the intervention period. This suggests that relatively low-cost, student-delivered MI may provide a pragmatic, sustainable approach to physical activity promotion in rural adults. Deeper, richer insights from the interviews shed light on the potential for the MI to challenge attitudinal barriers and limited physical activity “literacy” among this population, as well as the need to leverage community resources and social influences to support behavior change.

Accessibility of health services has been linked to the disparity in health status for rural residents [8]. An MI-based intervention delivered by final-year university placement students represents a feasible and cost-effective approach to delivering health promotion to relatively under-served rural communities. Recruitment for the intervention in the current study covered a region of over 900 square kilometers, despite the intervention being delivered from a single location in this region. All interview participants reported being satisfied with the location, with up to 20 km being identified as an acceptable distance to travel for a purpose such as this. More research is needed to confirm these findings in regions that differ in remoteness from major cities, to more confidently develop effective marketing strategies that maximize the awareness of, and recruitment into, the intervention across the full breadth of rural Australia.

Interview responses regarding motivation to exercise identified two groups: participants who exhibited self-efficacy but had no intention or lacked motivation to increase physical activity participation; and participants who were motivated or had intention but lacked self-efficacy. This categorization may help to inform MI content. The former would arguably benefit from more education to improve health literacy in relation to the benefits of regular physical activity, while the latter group may benefit from more practical counseling with a focus on prior accomplishments, facilitation of planning, overcoming barriers, and linking to resources [15].

Interview participants identified interpersonal factors that might limit engagement with the intervention, such as the feeling that they might be “holding others up”, or facilitate engagement, such as the presence of a local community member on the team delivering the intervention. These findings shed light on aspects of program delivery that should be considered for future planning. Discomfort associated with seeing other members of the community waiting to enter the process could be minimized by choosing venues with sufficient floor space and furniture to increase privacy and reduce awareness of others in the room, while increasing staff numbers would improve “flow” and reduce waiting times during the process, from health assessments to completion of the MI. If participants feel more comfortable interacting with team members who are perceived as a member of the community, recruitment of local volunteers onto delivery teams, supported with suitable training and quality control, could be considered for further development of the strategy. Residents of the same region are likely to be more familiar with local social, cultural, environmental, and health service-related barriers and enablers for adoption of regular physical activity and may therefore be particularly well equipped to facilitate contextualized planning of behavior change by delivering the MI themselves or “immersing” other interviewers on the team in the local context prior to MIs. Residents of communities may also develop greater awareness of, and confidence in, the intervention strategy if local volunteers are strategically identified through local media outlets. Local, trained volunteers may also facilitate ongoing contact with participants, in the time between visits from the university teams, thereby maintaining intervention “dose” and the momentum of behavior change. An example of the power of incorporating local volunteers into a mobile health delivery service in a vulnerable community is the Family Van, which provides a wide range of health services in disadvantaged suburbs of Boston, USA [20]. By carefully considering local social and environmental factors, the Family Van provides services in a welcoming, culturally competent, linguistically appropriate, non-judgmental manner using trained staff from the local community familiar with the challenges their consumers are facing [21].

Female interview participants in the current study expressed some discomfort in social settings such as weight loss groups, associated with embarrassment about their weight status. A cornerstone of the SDT is the concept of relatedness, which in this context refers to the extent to which people feel comfortable in, and connected to, the environment within which a behavior takes place [22]. Markland and Tobin [23] introduced the concepts of personal relatedness and social assimilation as separately functioning aspects of relatedness. Recent research has identified (lack of) social assimilation as a barrier to physical activity participation among rural women (but not men) who had previously experienced a cardiac event and had been prescribed cardiac rehabilitation [24]. Social assimilation among rural women may reflect limited exercise facilities and socially meaningful choices for women, particularly in sparsely populated and relatively poorly resourced communities. Notably, Quirk and colleagues [24] also reported an association of social assimilation and support from their spouse or domestic partner among women, offering some insight into a feasible mechanism through which to increase social assimilation among females by targeting sources of support in the family. Together, these observations suggest that the MI component of the intervention described in the current study needs to be gender-specific, with an emphasis on supporting women to openly discuss their health issues and perceived barriers to engagement with physical activity in their environment.

Participants in the current study reported that they receive inconsistent or inadequate lifestyle advice from otherwise trusted sources of health support (GP or practice nurse). Alarmingly, an Australian survey of physical activity specialists and sports medicine professionals identified that only 37% of respondents were able to correctly describe extant Australian physical activity guidelines and 35% reported explicitly using the guidelines regularly with their clients [25]. For optimal impact of MI in a health promotion context, there is a need to address conflicting information from interviewers and local health services providers that could reduce the motivation to adopt appropriate levels of physical activity. Accordingly, upscaling of the current intervention strategy should consider inclusion of community-level practices to increase physical activity “literacy” in the general population as well as professional development workshops for health professionals, with the aim of improving consistency of messaging and accurate translation of current national physical activity guidelines for health outcomes.

There is growing agreement that a more facilitative “choice architecture” at the community level is required for individuals to make and sustain positive health behavior choices [26]. This is particularly the case in Australian rural communities where health tends to be conceptualized as a “functional” attribute, such as the ability to perform work-related tasks, and preventative health behaviors are less valued [27]. Further, there is mounting evidence that rural regions differ in socio-cultural attributes that influence health-related behaviors, and that a “one size fits all” approach to health promotion in rural regions will lead to a less than optimal use of resources [22,25]. Accordingly, it is critical that strategies that aim to change community attitudes and health literacy are co-developed with local stakeholders and decision-makers [28].

### Strengths and Limitations

A strength of the current study is the relatively inexpensive intervention delivery, through the involvement of trained allied health placement students in the collection of data and delivery of the MI and follow-up messaging. Throughout Australia, enrollment in allied health university programs has grown steadily in recent years [29], ensuring an ongoing pool of students from which teams can be recruited to sustain the intervention into the future. Further, in the last decade, there has been an acknowledgment by the federal government of the challenges faced by remote and regional Australian communities in relation to reliable health care [30]. In Australia, the number of allied health professionals per head of population decreases with increasing remoteness [31]. Where medical and allied health students experience positive rural placements, they are more likely to consider establishing their professional career in a rural community [32]. By providing students with opportunities to develop their understanding of the rural context, communication skills, team-orientated professional practice, decision-making, adaptability, and resilience, the strategy reported here is well placed to equip students for rural practice and, in the long term, address the unmet health needs of rural Australians [30].

There are study limitations that are worthy of acknowledgment. Follow-up feasibility and acceptability questionnaires were not completed by 20% of the participants who completed the initial questionnaire. Responses of those in this group may have differed from those who completed the questionnaires at both time points, potentially resulting in different feasibility and acceptability scores than those reported. For instance, it is possible that those who did not attend the second assessment session were less satisfied with the quality of the delivered MI, thereby introducing a bias in the analysis. Participants who took part in the intervention, and therefore the study, may have had an existing interest in or awareness of health, specifically physical activity, and are therefore unlikely to be representative of rural populations in general. The socioeconomic profile of the target region is not representative of rural South Australia, with relatively high levels of education and employment in the Barossa region [33]. Interviews were conducted with eight participants and it is unlikely that saturation of data was reached. Consideration should also be given to trustworthiness of participants’ responses in both the questionnaires and interviews as social desirability may have biased responses. Efforts were made to mitigate this risk by making participants aware that the data were de-identified and were being collected for action research (i.e., their reflection and feedback would be used to inform future interventions).

## 5. Conclusions

The novel approach to physical activity promotion evaluated in this study, comprising a health age assessment and tailored MI, was well received by participants. However, while feasibility and acceptability were strongly endorsed, participants identified personal and structural barriers to behavior change. Further development of this and other physical activity promotion strategies in rural regions will depend on a deeper understanding of individuals’ and communities’ awareness, attitudes, and beliefs in relation to active lifestyles.

## Figures and Tables

**Table 1 ijerph-18-01308-t001:** Mean scores (and standard deviations) for health measures.

	Time Point 1	Time Point 2
Participants (*n*)	58	45
Variables		
weight (kg)	75.9 (14.3)	73.6 (14.1)
height (cm)	166.7 (8.6)	167.1 (8.5)
body mass index (kg·m^−2^)	27.3 (4.5)	26.3 (4.3)
waist girth (cm)	90.0 (13.1)	87.0 (12.6)
hip girth (cm)	102.6 (9.7)	102.9 (10.1)
* systolic blood pressure (mmHg)	139.6 (18.0)	126.3 (19.1)
* diastolic blood pressure (mmHg)	80.6 (9.0)	78.8 (9.6)
left grip strength (kg)	29.7 (12.0)	29.6 (11.0)
right grip strength (kg)	31.7 (12.1)	33.0 (10.9)
reaction time (millisec)	543 (146)	493 (79)
forced vital capacity (L)	2.53 (0.79)	2.88 (0.89)
balance score	3.1 (2.0)	3.8 (1.7)
total cholesterol (mmol·L^−1^)	5.2 (1.2)	5.0 (1.3)
* estimated health age (years)	64.4 (12.3)	57.7 (11.0)

Note: * Statistically significant difference between time points 1 and 2.

## Data Availability

Not applicable.

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
