# Peer review of "Student-Led Motivational Interviewing for Physical Activity Promotion among Rural Adults: A Feasibility and Acceptability Trial"

_ijerph, 2021, doi:10.3390/ijerph18031308_

Round 1

Reviewer 1 Report

The manuscript is an intervention study with the purpose to test feasibility and acceptability of a physical activity focused motivational interview and tailored follow-up text or telephone contacts that reinforced motivational interview content, in adults from a rural South Australian region. There is a high component of originality in the study and the text is well-written and structured. The theoretical basis is well founded and justifies the study. The objective is clearly defined. The methods used are sufficiently described, appropriate and consistent with the study proposal. The results are presented in an attractive manner, and the discussion of the findings show good prospects for the area of knowledge, which enhances the quality of the manuscript. However, follow up feasibility and acceptability questionnaires were not completed by 20% of the participants who completed the initial questionnaire. Thus, as it is explicit in the limitations of the study (lines 346-350), responses of those in this group may have differed from those who completed the questionnaires at both time points, potentially resulting in different feasibility and acceptability scores than those reported. In this regard, This is essential to strengthen the study's findings. Therefore, I suggest that the authors address this fact in more detail in the discussion of the results, and not just as a limitation. The references are current and pertinent to the theme of study. However, it is necessary to standardize the format for citing references in the text. At certain times the citations are by the author (for example: Dietz, Douglas & Brownson 2016) and at other times by numbers (for example: 1, 2, 3).

Reviewer 2 Report

Review "Student-led motivational interviewing for physical  activity promotion among rural adults: a feasibility  and acceptability trial" Thank you for inviting me to the review. Abstract - approx Introduction The introduction is written briefly and addresses issues relevant to the article. The authors have analyzed very well various publications on health behavior (including physical activity) of people living in the countryside. Please expand the phrase "our team" (line 53) - are they employees of the University of Soutch Australia? Are you about the authors of this article? Material and method - approx Interpretation assessment - approx Study Participants: The authors write: ... Interview participants were selected by purposive sampling to ensure a rich description of participant experience (line 96) and ... In 2018, residents of South Australia's Barossa region were recruited through strategically placed flyers and local print media (line 67-68) - my question - how exactly were people recruited? Leaflets, media, deliberate sample selection? Questionnaires - ok Interviews - approx Data Analysis - approx Results - ok Questionnaires - ok Interviews - approx Accessibility - approx Clarity and credibility of intervention - ca. Social influences - approx Prior knowledge and exposure to the concept of health age - approx Attitudes and beliefs - approx Motivation - ok Community and social influences - approx Knowledge of the benefits of physical activity - very interesting statements of the respondents, which talk about how important it is to practice ordinary physical activity ... Discussion - It seems very important to create an appropriate environment for the respondents and, as the authors noted, local interviewers should be invited to cooperate. I think it would be a good idea to point these people out to relevant social media, TV programs and industry literature, etc. Which will encourage and support them to be more physically active. Strengths and Limitations It seems very important ... By providing students with opportunities to develop their understanding of the rural context, communication skills, team-orientated professional practice, decision-making, adaptability and resilience, the strategy reported here is well placed to equip students for rural practice and , in the long term, address the unmet health needs of rural Australians ... - this is a very good example of involving medical students in their future work in the profession. As the authors noted, the study should be continued on a larger population to be more reliable. References: 1. Cleland et al. 2017 is without a number - it is stuck to the number 9 and has a wrong spelling. 2. Is Norton and Norton # 15? - no year. 3. Item no. 27 - no full name of the journal is available 4. Rank 33 - occurs twice a year (2012).

Reviewer 3 Report

The paper entitled "Student-led motivational interviewing for physical activity promotion among rural adults: a feasibility and acceptability trial" presents several major structural concerns that need reformulation. Also, shows some carelessness in its elaboration. There are different text formatting, confusion in the application of bibliographic reference norms (more than one used in the text), and errors in the references…
The paper structure needs revision. The introduction should be rewritten to provide more pertinent background information and explain exactly what the paper will address and why.
The methodology section should provide the reader with more and detailed information. For me, it seems confusing the sampling criterion. I am not sure that there is enough evidence to be able to understand that decision. There are no references that supported the methodological decisions, and vague information concerning the topics of the interview guide.
The data analysis procedure deserves improvement. Qualitative research is best at showing us specific instances of what might be general patterns of trends. It is better to focus on your data and what this means rather than being intent on extrapolating these findings prematurely (and when you do, you should be extremely cautious). Before revising this section it will be useful to read Johnny Saldaña work, namely some issues of his book (The Coding Manual for Qualitative Researchers) as ‘rising above the data’, ‘ordering and reordering’, ‘headings and subheadings’ and what he calls ‘buried treasure’. So, make the results section more substantive and address trustworthiness in methodology. Results need more interpretation, not only presents extracts of participants' interviews. The limitations are many and some could have been avoided. No detailed suggestions are pertinent before a manuscript reformulation. The manuscript as it stands needs extra work to become a more polished and focused piece.
